



**Effects of *a priori* profile shape assumptions on comparisons between satellite NO₂ columns**
**and model simulations**
Matthew J. Cooper[1,2], Randall V. Martin[2,1,3], Daven K. Henze[4], Dylan B.A. Jones[5]
1. Department of Physics and Atmospheric Science, Dalhousie University, Halifax, Nova Scotia,
Canada
2. Department of Energy, Environmental & Chemical Engineering, Washington University in St.
Louis, St. Louis, Missouri, USA
3. Harvard-Smithsonian Center for Astrophysics, Cambridge, Massachusetts, USA
4. Department of Mechanical Engineering, University of Colorado, Boulder, Colorado, USA
5. Department of Physics, University of Toronto, Toronto, Ontario, Canada
**Abstract**
A critical step in satellite retrievals of trace gas columns is the calculation of the air mass factor
(AMF) used to convert observed slant columns to vertical columns. This calculation requires *a*
*priori* information on the shape of the vertical profile. As a result, comparisons between satellite-
retrieved and model-simulated column abundances are influenced by the *a priori* profile shape.
We examine how differences between the shape of the simulated and *a priori* profile can impact
the interpretation of satellite retrievals by performing an adjoint-based 4D-Var assimilation of
synthetic NO₂ observations for constraining NOₓ emissions. We use the GEOS-Chem Adjoint
model to perform assimilations using a variety of AMFs to examine how *a posteriori* emission
estimates are affected if the AMF is calculated using an *a priori* shape factor that is inconsistent
with the simulated profile. In these tests, an inconsistent *a priori* shape factor increased errors in
*a posteriori* emissions estimates by up to 80% over polluted regions. As the difference between
the simulated profile shape and the *a priori* profile shape increases, so do the corresponding
assimilated emission errors. This reveals the importance of using simulated profile information
for AMF calculations when comparing that simulated output to satellite retrieved columns.

**1. Introduction**

Satellite observations provide a wealth of information on the abundance of trace gases in

the troposphere (Fishman et al., 2008). The next generation of satellite instruments, including the
upcoming geostationary constellation of TEMPO (Chance et al., 2013; Zoogman et al., 2017),



Sentinal-4 (Bazalgette Courrèges-Lacoste et al., 2011; Ingmann et al., 2012), and GEMS (Bak et
al., 2013; Kim, 2012), will provide information on NO₂ and other air quality relevant pollutants
on unprecedented spatial and temporal scales. Insight into processes that affect atmospheric
composition, including emissions (Streets et al., 2013), lifetimes (Fioletov et al., 2015; de Foy et
al., 2015; Laughner and Cohen, 2019), and deposition (Geddes and Martin, 2017; Kharol et al.,
2018) can be gained by interpreting this information with atmospheric chemistry models.

There are three main stages in retrieving trace gas abundances from ultraviolet and

visible solar backscatter radiance measurements: calculating a light-path "slant column" by
fitting observed spectra to known spectral signatures of trace gases, removing the stratospheric
portion of the column, and converting the slant column to a vertical column density using an air
mass factor (AMF). AMFs are calculated using a radiative transfer model and are a function of
viewing geometry, surface reflectance, clouds, and radiative transfer properties of the
atmosphere. AMF calculations also require an *a priori* estimate of the trace gas vertical profile
and are sensitive to the profile shape (Eskes and Boersma, 2003; Palmer et al., 2001).
Uncertainties in AMF calculations are the dominant source of uncertainty in satellite NO₂
retrievals over polluted regions (Boersma et al., 2007; Martin et al., 2002) largely due to
sensitivity to surface reflectance, clouds, aerosols, and *a priori* profile information (Lorente et
al., 2017).

Boersma et al. (2016) highlighted the issue of representativeness errors in comparing

model simulated values with UV-Vis satellite-retrieved columns. Vertical representativeness
errors arise from the satellite's altitude-dependent sensitivity due to atmospheric scattering and
can degrade the quality of model-measurement comparisons beyond errors that arise from either
modeling or measurements alone. A consistent accounting of the altitude-dependent sensitivity is
necessary to limit these errors.

Two common methods are used to account for vertical representativeness. In one method,

observed slant columns are converted to vertical columns using an air mass factor calculated
with scattering weights to represent instrument vertical sensitivity and shape factors to represent
the vertical profile (Palmer et al., 2001). Another commonly used method employs an AMF
provided with the retrieval to convert slant columns to vertical columns, and then applies an
averaging kernel to the simulated profile to resample the simulated profile in a manner that
mimics the satellite vertical sensitivity (Eskes and Boersma, 2003). In this method both the





averaging kernel and the retrieval AMF are calculated using an *a priori* $NO_2$ profile that may
have a different shape than the simulated profile, which may introduce errors in the observation-
simulation comparison (Zhu et al., 2016).

A common application of comparisons between satellite observed columns and model

simulations is to constrain $NO_x$ emissions (e.g. Ding et al., 2018; Ghude et al., 2013; Lamsal et
al., 2011; Martin et al., 2003; Vinken et al., 2014). One such approach is the use of four-
dimensional variational (4D-Var) data assimilation, which seeks to minimize a cost function that
accounts for the difference between simulated and retrieved values. As the cost function is a
difference between observed and simulated $NO_2$ columns, it is susceptible to vertical
representativeness errors resulting from inconsistent *a priori* vertical profile information. Studies
have shown that differences in retrieval processes between different $NO_2$ column products,
including differences in *a priori* profile shape, can propagate into errors of up to 50% in adjoint
inversions of $NO_x$ emissions (Qu et al., 2017). Studies have shown that shape factor errors can
impact emission estimates from other inversion methods as well (Laughner et al., 2016).

In this work we examine how *a priori* profile assumptions impact satellite-model

comparisons and use the GEOS-Chem adjoint as a case study to assess how this impact can
affect the interpretation of satellite observations. Section 2 provides the mathematical framework
for AMF calculations and satellite-model comparisons. Section 3 describes the adjoint model and
synthetic observations for the case study. Section 4 discusses the results.

**2. Mathematical frameworks**
**2.1 AMFs and averaging kernels**

The air mass factor translates the line-of-sight slant column abundances ($\Omega_s$) retrieved

from satellite observed radiances into vertical column abundances ($\Omega_v$). An air mass factor is the
ratio of $\Omega_s$ to $\Omega_v$ and depends on the atmospheric path as determined by geometry, $NO_2$ vertical
profile (***n***), surface reflectance, and radiative transfer properties of the atmosphere. Here we use
*M(**n**)* to represent an air mass factor derived using the vertical number density profile ***n***:

$$M(\boldsymbol{n}) = \frac{\Omega_s}{\Omega_v} \tag{1}$$




In the method described by Palmer et al. (2001), a radiative transfer model is used calculate
scattering weights $w(z)$ (also known as box air mass factors) which characterize the sensitivity of
backscattered radiance $I_B$ to the abundance of a trace gas at altitude $z$:

$$w(z) = -\frac{1}{M_g}\frac{\alpha_{a,z}}{\alpha_{eff}}\frac{\partial \ln (I_B)}{\partial \tau} \qquad (2)$$

where $\alpha_{a,z}$ is the temperature-dependent absorption cross section (m$^2$ molec$^{-1}$), $\alpha_{eff}$ is the effective
(weighted average) absorption cross section (m$^2$ molec$^{-1}$) and $\partial \tau$ is the incremental trace gas
optical depth. $M_G$ represents a geometric path correction accounting for the satellite viewing
geometry:

$$M_G = \sec \theta_o + \sec \theta \qquad (3)$$

where $\theta$ is the solar zenith angle and $\theta_o$ is the satellite viewing angle. This information is then
combined with an *a priori* NO$_2$ shape factor (i.e. normalized vertical profile)

$$\boldsymbol{S}(z) = \frac{\boldsymbol{n}(z)}{\Omega_v} \qquad (4)$$

typically calculated with an atmospheric chemistry model to provide an air mass factor via:

$$M(\boldsymbol{n}) = \int_0^{tropopause} \boldsymbol{w}(z)\boldsymbol{S}(z)dz \qquad (5)$$

where $\boldsymbol{S}(z)$ is calculated using vertical profile $\boldsymbol{n}(z)$. An attribute of the formulation of Palmer et
al. (2001) is the independence of atmospheric radiative transfer properties $w(z)$ and the vertical
trace gas profile $S(z)$. The AMF definition in Equation (1) combined with Eq. (4) indicates that a
slant column can be calculated from a known vertical profile via:

$$\Omega_s = \int_0^{tropopause} \boldsymbol{w}(z)\boldsymbol{n}(z)dz \qquad (6)$$


In an alternative formulation, the air mass factor is represented as part of an averaging

kernel.  As formulated by Rodgers and Connor (2003), the averaging kernel ($\boldsymbol{A}$) provides the
information needed to relate the retrieved quantity $\widehat{\boldsymbol{n}}$ to the true atmospheric profile $\boldsymbol{n}$:

$$\widehat{\boldsymbol{n}} - \boldsymbol{n_a} = \boldsymbol{A}(\boldsymbol{n} - \boldsymbol{n_a}) \qquad (7)$$

where $\boldsymbol{n_a}$ is an assumed *a priori* profile of number density. The elements of averaging kernel are





related to the scattering weights by:

$$A(z) = \frac{w(z)}{M(n_a)} \tag{8}$$

where $M(n_a)$ is an air mass factor calculated using *a priori* vertical profile information. It is
important to note that unlike scattering weights, averaging kernels depend on the *a priori*
assumed vertical profile shape.

It is possible to decouple the radiative transfer information from the assumed vertical

profile information in an averaging kernel by converting the supplied averaging kernels to
scattering weights via:

$$w(z) = \frac{A(z)M(n_a)}{M_G} \tag{9}$$

A lexicon is given in Table 1 as notation used to describe these treatments has varied

across the literature. We choose $M$ for air mass factor as a single letter is clearer in equations, $w$
for scattering weights to maintain the original formulation of Palmer et al. (2001), $n$ for number
density following IUPAC recommendations, and $\Omega$ for column densities as is common in
radiative transfer literature.

Figure 1 shows examples of typical shape factor, scattering weight, and averaging kernel

profiles for a range of atmospheric conditions. $NO_2$ shape factors have significant variability;
Shape factors peak near the surface in urban regions due to local pollution sources, but peak in
the upper troposphere in more remote regions due to lightning. The shape of a scattering weight
profile depends strongly on surface reflectance and cloud conditions. Sensitivity in the lower
troposphere increases over reflective surfaces. Clouds increase sensitivity above due to their
reflectance but shield the satellite from observing the atmosphere below. Averaging kernels have
similarities with scattering weights but depend on both the shape of the prior and the satellite
sensitivity. As AMF calculations are a convolution of the shape factor and the scattering weight
profiles, these shapes affect $NO_2$ retrievals. For these examples, the AMF for a clear sky
observation with surface reflectance of 0.01 can range from 0.7 in an urban region to 1.7 in a
remote region. This large difference demonstrates the importance of the assumed profile shape to
the retrieval process.

**2.2 Comparing satellite observations to simulated values**



The following section expresses mathematically how satellite-model comparisons are made
using various *a priori* profiles.

### 2.2.1 Using scattering weights

Following Palmer et al. (2001), a retrieved vertical column ($\widehat{\Omega}_{v,o}$) is estimated using an
observed slant column $\Omega_{s,o}$ and a simulation-based air mass factor $M(\boldsymbol{n_m})$, which can be
calculated with Eq. (5) using the model-simulated $NO_2$ profile ($\boldsymbol{n_m}$):

$$\widehat{\Omega}_{v,o} = \frac{\Omega_{s,o}}{M(\boldsymbol{n_m})} \tag{10}$$


The difference $\Delta_m$ between the estimated retrieved column and the model-simulated vertical
column ($\Omega_{v,m}$) is:

$$\Delta_m = \Omega_{v,m} - \widehat{\Omega}_{v,o} \tag{11}$$

$$\Delta_m = \left(\sum_0^{tropopause} n_m\right) - \frac{\Omega_{s,o}}{M(\boldsymbol{n_m})} \tag{12}$$

Equation (12) describes how this comparison is used in practice. However, we can rearrange this
expression in terms of model ($\Omega_{s,m}$) and observed ($\Omega_{s,o}$) slant columns using the definition of air
mass factor:

$$\Delta_m = \frac{\Omega_{s,m}}{M(\boldsymbol{n_m})} - \frac{\Omega_{s,o}}{M(\boldsymbol{n_m})} \tag{13}$$

$$\Delta_m = \frac{1}{M(\boldsymbol{n_m})}\left(\Omega_{s,m} - \Omega_{s,o}\right) \tag{14}$$


### 2.2.2 Using averaging kernels


Comparison of simulated and retrieved columns using the averaging kernel is described
by Eskes and Boersma (2003) and in the retrieval documentation in Boersma et al. (2011). The
averaging kernel is applied to the simulated profile in order to sample the simulated column in a
manner that reflects the retrieval sensitivity:

$$\widehat{\Omega}_{v,m} = \sum_0^{tropopause} A\boldsymbol{n_m} \tag{15}$$





The resampled simulated column is then compared to the retrieved vertical column ($\Omega_{v,o}$) using
the *a priori*-based air mass factor $M(n_a)$ supplied with the retrieval dataset:

$$\Delta_a = \widehat{\Omega}_{v,m} - \Omega_{v,o} \tag{16}$$

$$\Delta_a = \sum_0^{tropopause} An_m - \frac{\Omega_{s,o}}{M(n_a)} \tag{17}$$

Equation (17) describes how this method is used in practice. To facilitate the comparison with
Eq. (14), Eq. (17) can be rewritten using an alternative formulation relating averaging kernels to
scattering weights:

$$\Delta_a = \sum_0^{tropopause} \frac{wn_m}{M(n_a)} - \frac{\Omega_{s,o}}{M(n_a)} \tag{18}$$

$$\Delta_a = \frac{1}{M(n_a)} \left( \Omega_{s,m} - \Omega_{s,o} \right) \tag{19}$$


By comparing Eq. (14) to Eq. (19), it is evident that the underlying difference between the two
approaches is the choice of *a priori* profile information used to calculate the AMF, as the
averaging kernel method is not independent of *a priori* profile assumptions. This bias could be
addressed by replacing the *a priori* -based AMF in Eq. (18) with a simulation-based AMF using
the following relationship (Boersma et al., 2016; Lamsal et al., 2010):

$$M(n_m) = M(n_a) \frac{\sum An_a}{\sum n_m} \tag{20}$$

It should be noted that both the averaging kernel and scattering weight methods are

equivalent for comparisons that examine ratios of retrieved and modeled columns:

$$r_m = \frac{\widehat{\Omega_{v,o}}}{\Omega_{v,m}} = \frac{\Omega_{s,o} \big/ M(n_m)}{\sum n_m} = \frac{\Omega_{s,o}}{\sum n_m} \frac{\sum n_m}{\sum wn_m} = \frac{\Omega_{s,o}}{\sum wn_m} \tag{21}$$

$$r_a = \frac{\Omega_{v,o}}{\widehat{\Omega_{v,m}}} = \frac{\Omega_{s,o} \big/ M(n_a)}{\sum An_m} = \frac{\Omega_{s,o} \big/ M(n_a)}{\sum wn_m / M(n_a)} = \frac{\Omega_{s,o}}{\sum wn_m} \tag{22}$$


For ratios, both methods are dependent on geophysical assumptions used to calculate scattering
weights but are independent of *a priori* profile information.






## 3. Tools and Methodology

### 3.1 GEOS-Chem and its adjoint



The GEOS-Chem chemical transport model (www.geos-chem.org) is used to create
synthetic $NO_2$ observations and for their analysis. The GEOS-Chem version used here is version
35j of the GEOS-Chem Adjoint model. GEOS-Chem includes a detailed oxidant-aerosol
chemical mechanism (Bey et al., 2001; Park et al., 2004) and uses assimilated meteorological
fields from the Goddard Earth Observation System (GEOS-5), with 47 vertical levels up to 0.01
hPa and a horizontal resolution of 4°x5°. Global anthropogenic $NO_x$ emissions are provided by
the Emission Database for Global Atmospheric Research (EDGAR) inventory (Olivier et al.,
2005) with regional overwrites over North America (EPA/NEI99), Europe (EMEP), Canada
(CAC), Mexico (BRAVO, (Kuhns et al., 2005)), and East Asia (Streets et al., 2006). Other $NO_x$
sources include biomass burning (GFED2 (Van der Werf et al., 2010)), lightning (Murray et al.,
2012), and soils (Wang et al., 1998). This model has been used previously to constrain $NO_x$
emissions (Cooper et al., 2017; Henze et al., 2009; Qu et al., 2017, 2019; Xu et al., 2013; Zhang
et al., 2016).
The GEOS-Chem adjoint (Henze et al., 2007, 2009) is used here to perform a 4D-Var
data assimilation. The adjoint seeks to iteratively minimize a cost function generally defined by
the difference between satellite retrieved and simulated columns (Δ, from either Eq. (12) or Eq.

(17)):

$$J = \frac{1}{2}\Delta^T S_o^{-1}\Delta + \frac{1}{2}\gamma_R(E - E_a)^T S_E^{-1}(E - E_a) \tag{22}$$

where $E$ and $E_a$ are the *a posteriori* and *a priori* emissions, $S_o$ and $S_E$ are the retrieval and *a*
*priori* emission error covariance matrices, and $\gamma_R$ is a regularization parameter that allows for
weighting the cost function towards the retrieved columns or *a priori* emissions.

### 3.2 Experiment Outline


In this study we perform 4D-Var data assimilation experiments to infer surface $NO_x$
emissions using synthetic $NO_2$ observations. We use synthetic observations built from a known
emission inventory to provide a "truth" that can be used to evaluate the inversion results. To
demonstrate how *a priori* profile information can propagate in an assimilation, we use either the





model profile ($\Delta_m$, Eq. (12)) or an *a priori* profile ($\Delta_a$, Eq. (17)) in the cost function. For these
tests, we use one observation per hour per 4°x5° grid box for a period of two weeks in July 2010.
A one-week spin-up window at the start of each adjoint iteration is used to allow $NO_x$ to reach
steady state. Observation error covariances $S_o$ are described as a relative error of 30% of the slant
column density, plus an absolute error of $10^{15}$ molecules $cm^{-2}$, which is representative of typical
satellite retrieved $NO_2$ column uncertainties (Boersma et al., 2007; Martin et al., 2002). We omit
the *a priori* emissions constraint in the cost function (i.e. set $\gamma_R$=0) to isolate the impact of the
observations.

### 219     3.2.1 Synthetic observations

Synthetic observations are created using a GEOS-Chem simulation where random

Gaussian noise with a standard deviation of 5% is added to the anthropogenic $NO_x$ emissions. No
additional noise is added to the individual observations to isolate the impact of AMF errors
against- additional sources of uncertainty. Figure 2 shows the standard (*a priori*) anthropogenic
$NO_x$ emissions and the changes used to create the "true" emissions for the synthetic
observations.

Observations consist of synthetic slant columns ($\Omega_{s,o}$) created by applying scattering

weights to the synthetic vertical profiles using Eq. (6). To represent typical conditions, average
scattering weight profiles for each grid box are found by averaging scattering weights for OMI
observations during July 2010. OMI scattering weights are calculated using the LIDORT
radiative transfer model (Spurr, 2002) by providing LIDORT with the observation geometry of
the OMI observations and aerosol profiles from the GEOS-Chem base simulation.

### 233     3.2.2 Shape Factors

To test the impact of *a priori* profile information, five different tests are performed using

five different $NO_2$ profile shapes for AMF calculations:

• Case $SF_M$ : The GEOS-Chem model simulated profile ($\boldsymbol{n_m}$), updated at each iteration

of the adjoint run

• Case $SF_{prior}$: The *a priori* GEOS-Chem simulated profile, without updating.

• Case $SF_{n30}$: An *a priori* profile created by a GEOS-Chem simulation where global

anthropogenic $NO_x$ emissions were perturbed with random Gaussian noise with a





241 standard deviation of 30%. In cases where this results in negative emissions, a value

242 of zero is used.

243 • Case $SF_{trop}$: An *a priori* profile that assumes the $NO_2$ profile shape is uniform from

244  the surface to the tropopause (~200 hPa).

245 • Case $SF_{BL}$: An *a priori* profile that assumes the $NO_2$ profile shape is uniform from the

246  surface to the boundary layer (~800 hPa).

248 An advantage of using scattering weights and the simulated shape factor in a 4D-Var framework

249 is that it allows for the shape factor, and thus the AMF, to be updated at each iteration. When *a*

250 *priori* profiles from an external source are used it is not possible for them to update during the

251 inversion. The $SF_M$ and $SF_{prior}$ cases will test the impact that iterative updates to the AMF will

252 have on *a posteriori* estimates. The additional cases will test for the impact of using an averaging

253 kernel based on *a priori* profile assumptions that are inconsistent with the model. In practice,

254 averaging kernels and *a priori* profiles included in retrieval data sets are generally derived from

255 chemical transport models that have different physical processes, emissions, or spatial

256 resolutions. The $SF_{n30}$ test is representative of an inversion that uses *a priori* profile information

257 from a different chemical transport model with similar resolution but different emissions. The

258 $SF_{BL}$ and $SF_{trop}$ tests are extreme examples of using an *a priori* based on a coarser resolution

259 model, as both tests assume no spatial variability. The $SF_{BL}$ profile is representative of polluted

260 regions as indicated by the typical urban profile in Fig. 1, while the $SF_{trop}$ profile is

261 representative of a typical rural profile.

263 **4. Results**

264  Figure 3 shows root mean square errors (RMSE) for the *a posteriori* emissions estimated

265 by the 4D-Var assimilation tests. All tests successfully reduce the *a priori* emission error by an

266 order of magnitude or more. The $SF_M$ has the lowest RMSE indicating that it can best estimate

267 the "true" emissions. The next lowest RMSE is for the $SF_{prior}$ test, which uses the same initial

268 model shape factor but does not update during the adjoint iterations, followed by the $SF_{n30}$,

269 $SF_{trop}$, and $SF_{BL}$ tests.

270  Figure 4 shows maps of the difference in RMSE between the $SF_M$ test and the other tests.

271 The $SF_M$ test has a lower RMSE than the other tests in 65-72% grid boxes where the difference is


nonzero. Again, the $SF_{prior}$ test is closest to the $SF_M$ test with a mean absolute difference of $6 \times 10^6$
molec/cm$^2$/s, followed by $SF_{n30}$ ($7 \times 10^6$ molec/cm$^2$/s), $SF_{trop}$, ($13 \times 10^6$ molec/cm$^2$/s), and $SF_{BL}$
($16 \times 10^6$ molec/cm$^2$/s).

Table 2 summarizes additional error statistics focused on grid boxes with significant

emission sources. Errors in *a posteriori* emission estimates are correlated with the "true"
emissions in the $SF_{trop}$ and $SF_{n30}$ tests, indicating that these tests are not well constraining the
emissions. Differences between tests are more significant over polluted regions where AMF
errors are more influential; For example, in the regions with the highest NO$_x$ emissions, RMSE
values indicate $SF_M$ outperforms $SF_{n30}$ by 30% and $SF_{trop}$ by >80%. Another sign of adjoint
inversion quality is a low variance in errors. While the posterior error is reduced relative to the *a*
*priori* error in all tests, error standard deviations are 30% higher for $SF_{n30}$ and 90% higher for
$SF_{trop}$ compared to SFM. The global maximum error for the SFtrop test is 30% higher than the
$SF_M$ test. All metrics indicate that the $SF_M$ test best represents the "true" emissions.

**5. Discussion**

Accounting for the vertical profile dependence of satellite observations is essential to

accurately interpret those observations. This work examines how the choice of shape factor
affects differences between simulated and satellite-retrieved quantities in a 4D-Var assimilation
framework. Examination of the mathematical frameworks behind two common methods for
comparing simulated and retrieved columns highlights how the method introduced by Palmer et
al. (2001) facilitates separation of observation sensitivity (scattering weights) from the profile
shape (shape factor) enabling the model-retrieval comparison to be independent of *a priori*
profile assumptions.

In these case studies, vertical representativeness errors were best reduced by using a

shape factor that was consistent with the model simulation. This was especially true in polluted
regions where the AMF errors dominate observation uncertainties, as deviations between the
tests were largest in these regions. The further the shape factor deviated from the model state the
larger the inversion errors became, as indicated by Fig. 5. Comparing the $SF_M$ and $SF_{prior}$ tests
shows that allowing for shape factor to update during the iterative adjoint process further reduces
the RMSE by 10%. However, even without allowing for shape factor updates, using a shape
factor that is consistent with the model state produces a more accurate inversion result than using



other assumed profile shapes.
The case study presented here demonstrates that the shape factor source can have a strong
influence on adjoint inversion results. However, the magnitude of this influence can vary.
Additional tests performed using synthetic observations built using random 15% or 30%
perturbations to emissions (rather than the 5% perturbation used here) were insensitive to the
AMF. In these tests, the adjoint cost function was more sensitive to the larger difference between
the observed and simulated slant columns (i.e. $\Omega_m$ - $\Omega_o$ in Eq. (13) and (18)) than to AMF. This
indicates that the adjoint inversion is less sensitive to vertical representativeness errors in cases
where emissions are poorly constrained; Conversely, choice of AMF will become increasingly
important to adjoint inversions as emission inventories improve. Furthermore, omitting the *a*
*priori* emissions constraint in the cost function and omitting noise in the observations in these
tests to isolate the impact of the AMF effectively assumes poorly constrained *a priori* emissions
and ideal observations. In practice, cost function sensitivity to AMF choice may be buffered
when *a priori* emissions uncertainties and observational noise are considered.
As it is beneficial for a consistent shape factor to be used when comparing satellite
retrieved values to model simulated results, it will be useful for data products to provide the
information required for this method to the user community. This is most straightforward when
scattering weights (rather than averaging kernels) are provided alongside retrieved column data,
as scattering weights and shape factors are independently calculated, however averaging kernels
can be converted to scattering weights if the *a priori* profiles used are included in the dataset.
In summary, when comparing a model simulation to a satellite retrieved $NO_2$ column in a
4D-Var environment, calculating the AMF using the simulated shape factor allows for better
accuracy in inversion results. This demonstration can provide general guidance for other
methods of interpreting satellite observations with models, as using the simulated shape factor
assures consistency in the vertical representativeness between model and retrieval.

**7. Author Contributions**
MC designed and carried out the case studies and their analysis. All co-authors provided
guidance in analyzing results. MC prepared the manuscript with contributions from all co-
authors.



**8. Competing interests**

The authors declare that they have no conflict of interest.

**9. Acknowledgements**

This work was supported by the Canadian Space Agency. DH acknowledges support from
NASA NNX17AF63G.

**10. Data Availability**

The GEOS-Chem chemical transport model and its adjoint are available at www.geos-chem.org
(last access: 20 August 2017). OMI NO$_2$ data used in this study is available from the NASA
Goddard Earth Sciences Data and Information Services Center (https://disc.sci.gsfc.nasa.gov;
last access: 14 March 2019). AMF code (Spurr, 2002; Martin et al., 2002) used to calculate
scattering weights and air mass factors is available at http://fizz.phys.dal.ca/~atmos (last access:
19 June 2017).

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





Figures:

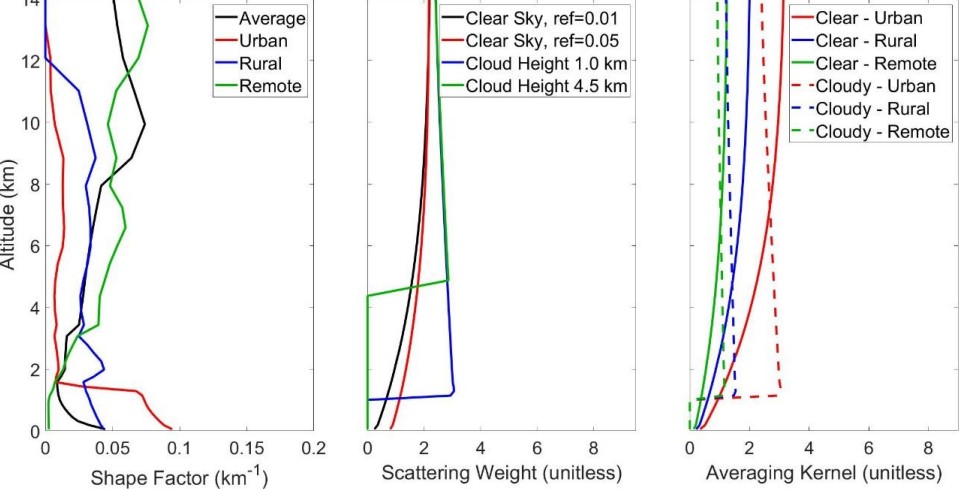


Figure 1: (Left) Shape factor profiles from a GEOS-Chem simulation for July 2010. Shown are a
global average, and typical urban (Beijing), rural (Midwest USA), and remote (Tropical Pacific)
profiles. (Middle) Typical OMI scattering weight profiles for varying surface reflectance and
cloud height. (Right) Averaging kernels calculated using the same shape factors and scattering
weights ("Clear Sky" surface reflectance is 0.01, "Cloudy" uses cloud height of 1 km).

**Anthropogenic NOₓ Emissions**

$NO_x$ Emissions (molec cm$^{-2}$ s$^{-1}$)

**Emission Perturbation**

Truth/a priori Emission Ratio (unitless)

Figure 2 (top) Anthropogenic NOₓ emissions for July 2010 used in GEOS-Chem. (bottom) Ratio
of "true" emissions used to create synthetic observations to a priori NOₓ emissions.



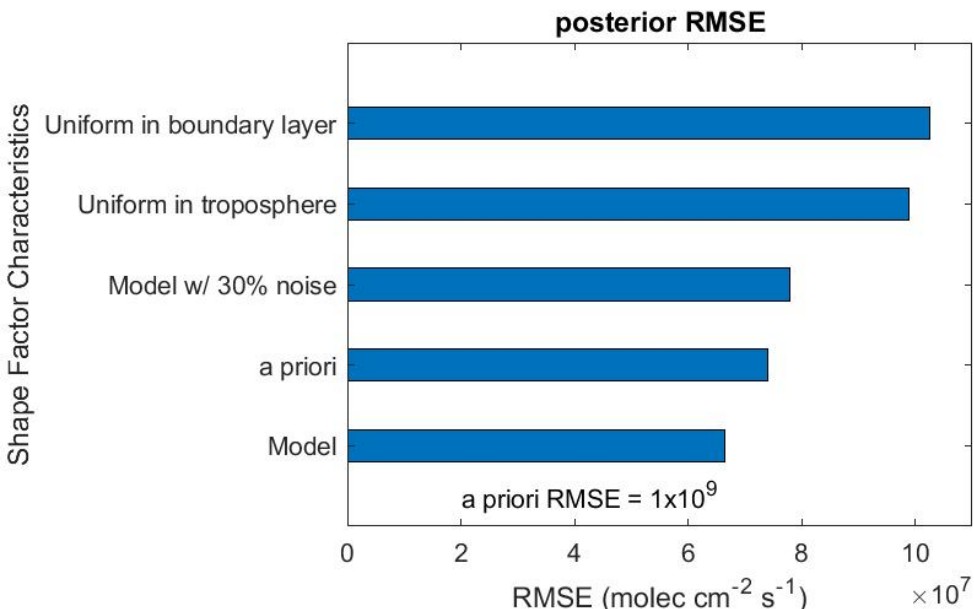

Figure 3: Global root mean square error (RMSE) values for 4D-Var estimates of $NO_x$ emissions
for tests using various shape factors in AMF calculations.

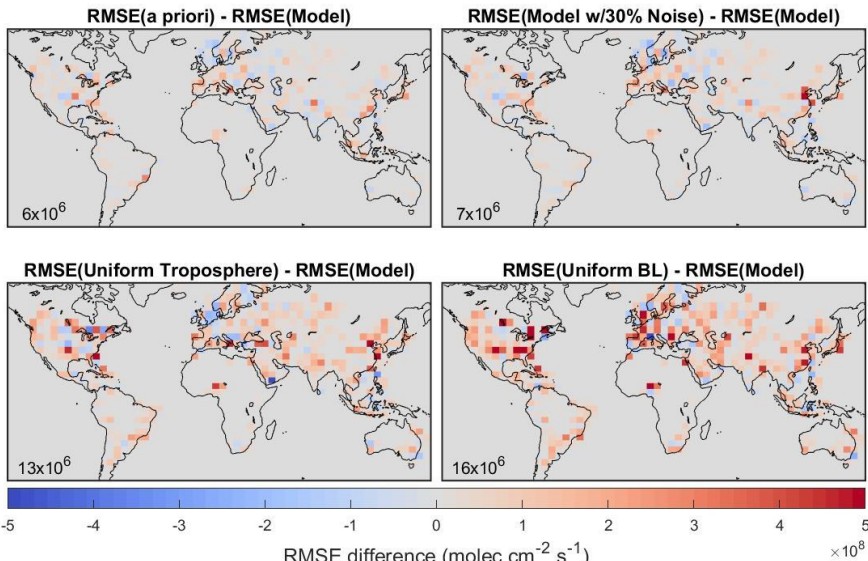

Figure 4: Difference between root mean square error (RMSE) of adjoint tests. Mean absolute
difference (molec/cm$^2$/s) values inset.



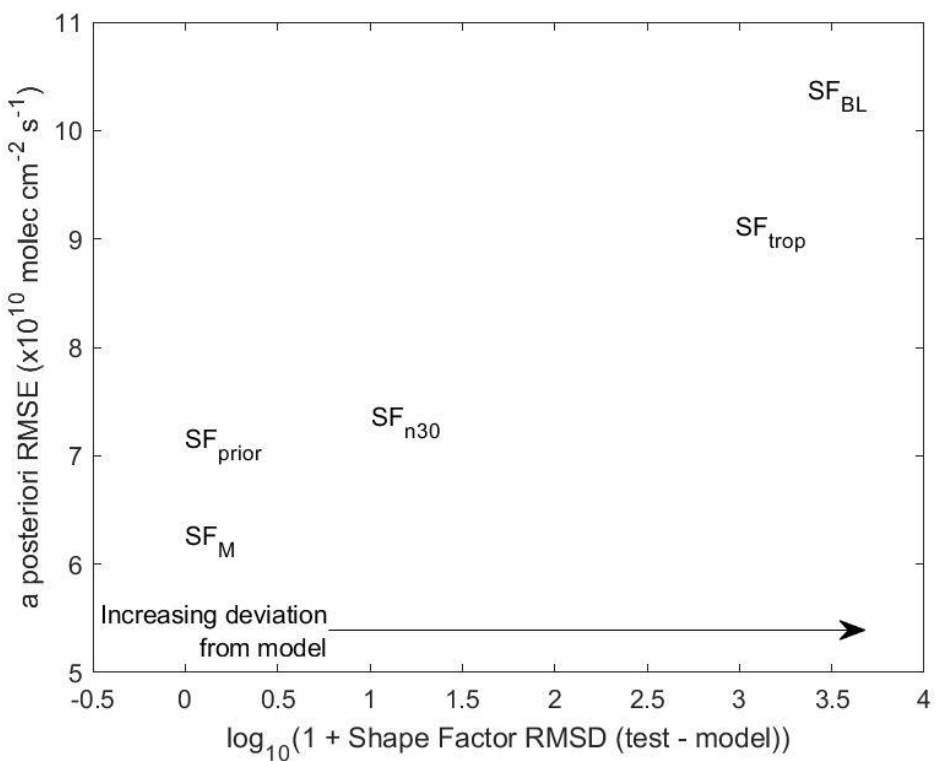

Figure 5: Scatterplot of adjoint test results. X-axis represents the deviation of the shape factor
from the model simulated shape factor (root mean square difference). Y-axis represents the *a*
*posteriori* emissions error from the adjoint inversion.



| Variable | *Palmer et al., 2001* | *Eskes & Boersma, 2003* | *Boersma et al., 2016* | Notation used here |
|---|---|---|---|---|
| Air mass factor | AMF | M | M | M |
| Slant Column | $\Omega_S$ | S | $N_S$ | $\Omega_s$ |
| Vertical Column | $\Omega_V$ | V | $N_V$ | $\Omega_v$ |
| Scattering Weight | $w(z)$ | $C_l$ | $m_l$ | w |
| Shape Factor | $S_z(z)$ | | | $S(z)$ |
| Averaging Kernel | | A | A | A |
| Number density | $n(z)$ | x | $x_l$ | $n(z)$ |
| Geometric AMF | $AMF_G$ | | | $M_G$ |

Table 1: Lexicon comparing notation used in this paper to that used in previous studies.






| Test Name | Shape Factor Source | Correlation (r) of *a posteriori* RMSE and "true" emissions | *a posteriori* RMSE ($\times 10^8$ molec/cm$^2$/s) | | Error standard deviation ($\times 10^8$ molec/cm$^2$/s) | | Maximum error ($\times 10^9$ molec/cm$^2$/s) |
|---|---|---|---|---|---|---|---|
| | | if "true" emissions $> 10^{10}$ molec/cm$^2$/s | "true" emissions $> 10^{10}$ molec/cm$^2$/s | "true" emissions $> 10^{11}$ molec/cm$^2$/s | "true" emissions $> 10^{10}$ molec/cm$^2$/s | "true" emissions $> 10^{11}$ molec/cm$^2$/s | |
| SF$_M$ | Model | 0.03* | 1.8 | 3.0 | 1.8 | 2.9 | 1.6 |
| SF$_{prior}$ | a priori | 0.03* | 2.0 | 3.2 | 2.0 | 3.3 | 1.6 |
| SF$_{n30}$ | Model w/ 30% noise | 0.16 | 2.1 | 3.9 | 2.1 | 3.8 | 1.8 |
| SF$_{trop}$ | Uniform in troposphere | 0.68 | 2.8 | 5.6 | 2.8 | 5.5 | 2.1 |
| SF$_{BL}$ | Uniform in boundary layer | 0.08* | 2.8 | 4.6 | 2.8 | 4.6 | 1.9 |

Table 2: Summary of error statistics for adjoint tests. Values marked * indicate that correlation is
not statistically significant (p>0.05)