# Peer review of "Effects of *a priori* profile shape assumptions on comparisons between satellite NO2 columns and model simulations"

_Atmospheric Chemistry and Physics, 2020_

## Referee Comment (RC1) · Anonymous Referee #1 · 25 Mar 2020

**March 25, 2020**

Cooper et al. "Effects of *a priori* shape assumptions on comparisons between satellite $NO_2$ columns and model simulations" presents a case study using synthetic OMI observations and the GEOS-Chem adjoint to show how different methods of accounting for the vertical sensitivity of satellite $NO_2$ measurements when comparing to model $NO_2$ fields affects emissions inferred from said comparison. This paper may have been a better fit for GMD rather than ACP, as it primarily touches on model comparison, but is also relevant to the remote sensing community as it informs what information must be contained in satellite products for effective model comparison, and to the broader atmospheric community for understanding possible sources of error in constrained emissions, so ACP is also appropriate.

I do have two concerns about the experimental design. First, I question whether using scattering weights computed for average OMI observing geometry in a $4° \times 5°$ grid cell is appropriate for creating synthetic observations. Second, some of the choices for prior test cases are not very relevant to current $NO_2$ retrievals. I will address these more below. If the authors can address these concerns, then this manuscript should be published in ACP.

**Major concerns**

**Representativeness of average scattering weights**

In sect. 3.2.1, line 226, the authors say: "To represent typical conditions, average scattering weight profiles for each grid box are found by averaging scattering weights for OMI observations during July 2010." I have two questions about this. First, it is ambiguous whether the mean scattering weights in question are found by averaging weights providing in the product (implied by "...average scattering weight profiles for each grid box are found by averaging scattering weights for OMI observations during July 2010...") or by using the average viewing conditions to compute the scattering weight vector for the average viewing angles, albedo, surface pressure, etc. (implied by "...OMI scattering weights are calculated using the LIDORT radiative transfer model (Spurr, 2002) by providing LIDORT with the observation geometry of the OMI observations and aerosol profiles from the GEOS-Chem base simulation..."). I'm assuming the latter, but this could be made clearer.

Assuming that the authors calculated their own scattering weights from the average viewing conditions, my second concern is that while this simplifies the problem of computing synthetic observations for each model step, it may not adequately represent the variation in

OMI measurements during that time. Scattering weights depend nonlinearly on observation geometry, so:

$$w(z|\overline{\theta}_s, \overline{\theta}_v, \overline{\phi}, \overline{a}_{\text{surf}}, \overline{p}_{\text{surf}}) \neq \frac{1}{c_{\text{obs}}} \sum_{i=1}^{c_{\text{obs}}} w(z|\theta_{s,i}, \theta_{v,i}, \phi_i, a_{\text{surf},i}, p_{\text{surf},i})$$

where $\theta_s$ is the solar zenith angle, $\theta_v$ the viewing zenith angle, $\phi$ the relative azimuth angle, $a$ the surface reflectivity, $p_{\text{surf}}$ the surface pressure, and overlined quantities represent grid cell averages. In other words, the vector of scattering weights corresponding to the average observation conditions is not guaranteed to be the same as the average of scattering weight vectors for all individual observations. Concretely, consider two observations, one with a viewing zenith angle of 0° and one at $\sim 60°$. The average of these two observations' scattering weights is not guaranteed to be the same as for an observation of 30°.

That being said, it may well be close enough, especially averaged over a $4° \times 5°$ grid cell. If the authors can show that the difference between using mean scattering weights and the mean of synthetic observations computed using individual OMI observation scattering weights is within the measurements' uncertainty for at least a few days of synthetic observations, then I think that would be adequate.

**Relevance of prior test cases**

Of the shape factor test cases described in sect. 3.2.2, the $SF_{trop}$ and $SF_{BL}$ cases are not particularly relevant for satellite measurements. Both of the two main global OMI $NO_2$ retrievals (NASA SP3, Krotkov et al. 2017; QA4ECV, see Williams et al. 2016 for the $NO_2$ profiles) use $\sim 1°$ resolution for their $NO_2$ profiles. Therefore, the $SF_{BL}$ and $SF_{trop}$ cases, which assume one profile globally are not representative of any major satellite product. In fact, the more relevant question, assuming that $4° \times 5°$ is still a common resolution for adjoint modeling, is what happens if the satellite prior is *higher* resolution than the model profile.

In my opinion, a sixth test case similar to $SF_{prior}$ but using a set of priors from a $2° \times 2.5°$ or $1° \times 1.25°$ GEOS-Chem simulation would add value to the paper by studying the effect of having the satellite product's prior at higher spatial resolution than the adjoint model. Also, if the $SF_{BL}$ and $SF_{trop}$ cases are retained, it should be clearly stated that they represent extreme cases that do not represent any modern $NO_2$ product.

**Other primary concerns**

- In sect. 3.2, line 210, the authors say that they use one observation per grid box per hour. But OMI will only observe a given location twice per day, maximum, and usually only once per day at about 13:30 local standard time. Are you then filtering these once-per-hour observations down to the ones OMI would actually observe?

- I don't follow Eq. (20). Specifically why $\mathbf{n_a}$ shows up on the right hand side. Given that:

$$M(\mathbf{n}) = \frac{\sum_i \mathbf{w}_i \mathbf{n}_i}{\sum_i \mathbf{n}_i}$$

and

$$\mathbf{A}_i(\mathbf{n}) = \frac{\mathbf{w}_i}{M(\mathbf{n})}$$

then to compute $M(\mathbf{n}_m)$ given $\mathbf{A}(\mathbf{n}_a)$ you only need to multiply $\mathbf{A}(\mathbf{n}_a)$ by $M(\mathbf{n}_a)$ to extract the necessary scattering weights:

$$M(\mathbf{n}_m) = \frac{\sum_i \mathbf{w}_i \mathbf{n}_{m,i}}{\sum_i \mathbf{n}_{m,i}}$$
$$\mathbf{w}_i = \mathbf{A}_i(\mathbf{n}_a) \cdot M(\mathbf{n}_a)$$
$$\therefore M(\mathbf{n}_m) = \frac{\sum_i \mathbf{A}_i(\mathbf{n}_a) M(\mathbf{n}_a) \mathbf{n}_{m,i}}{\sum_i \mathbf{n}_{m,i}}$$
$$= M(\mathbf{n}_a) \frac{\sum_i \mathbf{A}_i(\mathbf{n}_a) \mathbf{n}_{m,i}}{\sum_i \mathbf{n}_{m,i}}$$

I don't think you need $\mathbf{n}_a$ to compute $M(\mathbf{n}_m)$ as long as $M(\mathbf{n}_a)$ is included in the satellite data (both the NASA SP3 and QA4ECV OMI $NO_2$ products include the tropospheric AMFs), and given only the AKs and prior profile, it would be difficult if not impossible to compute $M(n_a)$. That means the statement on line 320 about needing the a priori profiles in the dataset is incorrect.

**Minor corrections**

- For Eq. (8), it would be good to make clear that $\mathbf{A}(z)$ is the column averaging kernel and therefore a vector, since in Rodgers and Conner, the capital $\mathbf{A}$ is typically the full AK matrix. But I agree that following the convention in Eskes and Boersma (2003) is best.

- Eq. (9) doesn't seem to be used anywhere else in the paper, and technically is inconsistent with the implicit definition of $\mathbf{w}(z)$ in Eq (8). Recommend removing Eq. (9).

- In Eq. (17) and Eq. (18) it's unclear what is being summed. Recommend using $i$ subscripts to make clear what terms are iterated in the sum.

- On line 198 in sect. 3.1, the authors say that $\Delta$ is computed using either Eq. (12) or (17). Given that much of sect. 2 was spent establishing that these two equations differ, this should be clarified. If I understood correctly, which equation is used effectively depends on which shape factor was used for a given test. If so, I recommend saying that explicitly.

- For the different shape factors, have you considered the impact of profiles simulated by a model with systematically, rather than randomly (as in $SF_{n30}$), different emissions?

**References**

Eskes, H. J. and Boersma, K. F.: Averaging kernels for DOAS total-column satellite retrievals, Atmospheric Chemistry and Physics, 3, 1285–1291, doi:10.5194/acp-3-1285-2003, URL https://www.atmos-chem-phys.net/3/1285/2003/, 2003.

Krotkov, N. A., Lamsal, L. N., Celarier, E. A., Swartz, W. H., Marchenko, S. V., Bucsela, E. J., Chan, K. L., Wenig, M., and Zara, M.: The version 3 OMI NO2 standard product, Atmospheric Measurement Techniques, 10, 3133–3149, doi:10.5194/amt-10-3133-2017, URL https://doi.org/10.5194/amt-10-3133-2017, 2017.

Williams, J. E., Boersma, K. F., Sager, P. L., and Verstraeten, W. W.: The high-resolution version of TM5-MP for optimised satellite retrievals: Description and Validation, doi: 10.5194/gmd-2016-125, URL https://doi.org/10.5194/gmd-2016-125, 2016.

---

## Referee Comment (RC2) · Anonymous Referee #2 · 5 Apr 2020

This manuscript addresses an interesting feature of sensitivity of NOx emission inversions to a priori profile shape assumptions in AMF of satellite NO2 columns. Authors conclude "As the difference between the simulated profile shape and the a priori profile shape increases, so do the corresponding assimilated emission errors". In the discussion section, however, the authors indicate that the adjoint inversion is less sensitive to vertical representativeness errors in cases where emissions are poorly constrained. It is noted that choice of AMF will become increasingly important to adjoint inversions as emission inventories improve. The manuscript delivers some new and intriguing messages to satellite and air quality modeling community. I think the manuscript is well written. The introduction and revisit of AMF and averaging kernel are neat and

helpful. There are some parts that need further investigation and explanation. Hope the authors revise and improve the manuscript before final publication.

* Major point Line 304-310: the manuscript deals with the "truth" emissions that deviate only 5% from the original anthropogenic NOx emissions. Here, it is written that the tests using random 15% or 30% perturbations to emissions were insensitive to the AMF. In real cases, NOx emission inventory errors are quite large (> 30%). Do the authors mean that choice of a priori shape factor is not important for most of real emission study cases? Please show the results from the random 15% or 30% perturbation tests (or other new cases if possible) and discuss more on applications to the real world problems (e.g., Qu et al., 2017).

* Minor points 1. Examples of inconsistent a priori shape factor: I do not think these days retrieval groups use SF_BL, SF_Trop type a piori. In the abstract, up to 80% increased error is based on this choice. I am not sure if readers need to take this number seriously.

2. Line 274-276: I am not sure what these mean.

3. Line 288-292: Examination of the mathematical frameworks behind two common methods for comparing simulated and retrieved columns highlights how the method introduced by Palmer et al. (2001) facilitates separation of observations sensitivity (scattering weights) from the profile shape (shape factor) enabling the model-retrieval comparison to be independent of a priori profile assumptions. In the last part, model-retrieval comparison to be independent of a priori. . .It is confusing because the main conclusion of the manuscript is that the model-retrieval comparison is not independent of a priori (in certain cases).

4. Line 307: Add "s" in the subscript.

5. Line 315-Line 320: It is good to emphasize these again. But I believe that retrieval groups are already doing this. It might be good to mention various data supported by

the retrieval groups.

6. Can the posteriori NOx emission difference (%) for highly polluted cases be shown in Table 2? This type of results will also be useful for higher perturbation cases.

7. Is the number of iterations of 4D VAR assimilations for all the test cases the same? How many iterations are requited for the tests?

8. Is there a possibility that model spatial resolutions affect the results?

---

## Author Comment (AC2) · 13 May 2020

**This manuscript addresses an interesting feature of sensitivity of NOx emission inversions to a priori profile shape assumptions in AMF of satellite NO2 columns. Authors conclude "As the difference between the simulated profile shape and the a priori profile shape increases, so do the corresponding assimilated emission errors". In the discussion section, however, the authors indicate that the adjoint inversion is less sensitive to vertical representativeness errors in cases where emissions are poorly constrained. It is noted that choice of AMF will become increasingly important to adjoint inversions as emission inventories improve. The manuscript delivers some new and intriguing messages to satellite and air quality modeling community. I think the manuscript is well written. The introduction and revisit of AMF and averaging kernel are neat and helpful. There are some parts that need further investigation and explanation. Hope the authors revise and improve the manuscript before final publication.**

Thank you for your comments. We address the specific comments below.

**Major point Line 304-310: the manuscript deals with the "truth" emissions that deviate only 5% from the original anthropogenic NOx emissions. Here, it is written that the tests using random 15% or 30% perturbations to emissions were insensitive to the AMF. In real cases, NOx emission inventory errors are quite large (> 30%). Do the authors mean that choice of a priori shape factor is not important for most of real emission study cases? Please show the results from the random 15% or 30% perturbation tests (or other new cases if possible) and discuss more on applications to the real world problems (e.g., Qu et al., 2017).**

Results from the random 30% perturbation tests are now included in the Results section.

We now clarify in the conclusion that our results show that recent emission study cases are likely insensitive to the AMF (Line 330):
"This indicates that while the adjoint cost function is mathematically dependent on the AMF, the inversion is less sensitive to vertical representativeness errors in cases where emissions are poorly constrained, as is the case in recent adjoint inversion studies (e.g. Qu et al., 2017). However, choice of AMF will become increasingly important to adjoint inversions as emission inventories improve."

**Minor points 1. Examples of inconsistent a priori shape factor: I do not think these days retrieval groups use SF_BL, SF_Trop type a piori. In the abstract, up to 80% increased error is based on this choice. I am not sure if readers need to take this number seriously.**

We clarify in Section 3.2.2 that the SFBL and SFtrop tests are extreme cases that do not represent typical retrievals (Line 264):

"The $SF_{BL}$ and $SF_{trop}$ tests do not represent any modern retrieval algorithms, but are used as extreme examples of an *a priori* that assumes no spatial variability."

We also changed the value cited in the abstract to 30%, representing the error of the more realistic test cases.

**2. Line 274-276: I am not sure what these mean.**

We have replaced these statistics with root mean square differences, which more clearly describe the difference between the *a posteriori* emissions from the $SF_M$ and the other tests.

**3. Line 288-292: Examination of the mathematical frameworks behind two common methods for comparing simulated and retrieved columns highlights how the method introduced by Palmer et al. (2001) facilitates separation of observations sensitivity (scattering weights) from the profile shape (shape factor) enabling the model-retrieval comparison to be independent of a priori profile assumptions. In the last part, model-retrieval comparison to be independent of a priori. . .It is confusing because the main conclusion of the manuscript is that the model-retrieval comparison is not independent of a priori (in certain cases).**

The mathematical framework outlined in Section 2 shows that indeed any model-retrieval comparisons is dependent on the prior. One conclusion in the manuscript is that the 4D-Var assimilation process, while mathematically dependent on the prior, is not sensitive to the prior under some conditions. We have clarified this by adding Table 2 which indicates how the model-retrieval comparisons are affected by the *a priori* profile, and by adjusting the discussion to clarify this distinction (Line 326):

"Inversion tests performed using synthetic observations based on random 30% perturbations to emissions were insensitive to the AMF, despite large differences in *a priori* vertical column densities. In these tests, the adjoint cost function was more sensitive to the larger difference between the observed and simulated slant columns (i.e. $\Omega_{s,m}$ - $\Omega_{s,o}$ in Eq. (13) and (19)) than to AMF. This indicates that while the adjoint cost function is mathematically dependent on the AMF, the inversion is less sensitive to vertical representativeness errors in cases where emissions are poorly constrained…"

**4. Line 307: Add "s" in the subscript.**

This has been corrected.

**5. Line 315-Line 320: It is good to emphasize these again. But I believe that retrieval groups are already doing this. It might be good to mention various data supported by the retrieval groups.**

Some retrieval data products do include scattering weights, but not all of them. However, we feel it is best to emphasize this as a general recommendation without singling out any particular retrieval groups.

**6. Can the posteriori NOx emission difference (%) for highly polluted cases be shown in Table 2? This type of results will also be useful for higher perturbation cases.**

Percent differences were not included in Table 2 for clarity. However, we now include mean "true" emission values in the captions to allow the reader to make such comparisons if desired.

**7. Is the number of iterations of 4D VAR assimilations for all the test cases the same? How many iterations are requited for the tests?**

We now mention the number of 4D Var iterations on Line 198: "Tests performed here required 20-30 iterations to minimize the cost function."

**8. Is there a possibility that model spatial resolutions affect the results?**

The mathematical framework outlined in Section 2 presents the main point of the paper, which is that consistency between simulated profiles and shape factor profiles used to calculate AMFs is essential. This is true regardless of model resolution. We now include a test where the *a priori* profile is generated by a higher resolution model (GEOS-Chem run at 2x2.5 resolution). This test further supports the main conclusion of the paper. From Line 316:

"The $SF_{finer}$ test indicates that using a higher resolution model to generate *a priori* profiles does not provide an advantage in simulation-observation comparisons, as consistency between the simulation profile and the AMF shape factor is of greater importance."

---

## Author Response (AR1)

**Reviewer 1 Comments and Responses:**

Cooper et al. "Effects of *a priori* shape assumptions on comparisons between satellite NO₂ columns and model simulations" presents a case study using synthetic OMI observations and the GEOS-Chem adjoint to show how different methods of accounting for the vertical sensitivity of satellite NO₂ measurements when comparing to model NO₂ fields affects emissions inferred from said comparison. This paper may have been a better fit for GMD rather than ACP, as it primarily touches on model comparison, but is also relevant to the remote sensing community as it informs what information must be contained in satellite products for effective model comparison, and to the broader atmospheric community for understanding possible sources of error in constrained emissions, so ACP is also appropriate.

   I do have two concerns about the experimental design. First, I question whether using scattering weights computed for average OMI observing geometry in a $4° \times 5°$ grid cell is appropriate for creating synthetic observations. Second, some of the choices for prior test cases are not very relevant to current NO₂ retrievals. I will address these more below. If the authors can address these concerns, then this manuscript should be published in ACP.

**Major concerns**

**Representativeness of average scattering weights**

In sect. 3.2.1, line 226, the authors say: "To represent typical conditions, average scattering weight profiles for each grid box are found by averaging scattering weights for OMI observations during July 2010." I have two questions about this. First, it is ambiguous whether the mean scattering weights in question are found by averaging weights providing in the product (implied by "...average scattering weight profiles for each grid box are found by averaging scattering weights for OMI observations during July 2010...") or by using the average viewing conditions to compute the scattering weight vector for the average viewing angles, albedo, surface pressure, etc. (implied by "...OMI scattering weights are calculated using the LIDORT radiative transfer model (Spurr, 2002) by providing LIDORT with the observation geometry of the OMI observations and aerosol profiles from the GEOS-Chem base simulation..."). I'm assuming the latter, but this could be made clearer.

   Assuming that the authors calculated their own scattering weights from the average viewing conditions, my second concern is that while this simplifies the problem of computing synthetic observations for each model step, it may not adequately represent the variation in OMI measurements during that time. Scattering weights depend nonlinearly on observation geometry, so:

$$w(z|\overline{\theta}_s, \overline{\theta}_v, \overline{\phi}, \overline{a}_{\mathrm{surf}}, \overline{p}_{\mathrm{surf}}) \neq \frac{1}{c_{\mathrm{obs}}} \sum_{i=1}^{c_{\mathrm{obs}}} w(z|\theta_{s,i}, \theta_{v,i}, \phi_i, a_{\mathrm{surf},i}, p_{\mathrm{surf},i})$$

where $\theta_s$ is the solar zenith angle, $\theta_v$ the viewing zenith angle, $\varphi$ the relative azimuth angle, $a$ the surface reflectivity, $p_{surf}$ the surface pressure, and overlined quantities represent grid cell averages. In other words, the vector of scattering weights corresponding to the average observation conditions is not guaranteed to be the same as the average of scattering weight vectors for all individual observations. Concretely, consider two observations, one with a viewing zenith angle of $0°$ and one at $\sim 60°$. The average of these two observations' scattering weights is not guaranteed to be the same as for an observation of $30°$.

That being said, it may well be close enough, especially averaged over a $4°{\times}5°$ grid cell. If the authors can show that the difference between using mean scattering weights and the mean of synthetic observations computed using individual OMI observation scattering weights is within the measurements' uncertainty for at least a few days of synthetic observations, then I think that would be adequate.

Thank you for your comments. The method we use to calculate average scattering weight profiles is to provide LIDORT with observation geometry from individual OMI observations, and then average the resulting scattering weight profiles. We have edited the text to clarify this procedure (Line 224):

"Scattering weights are calculated using the LIDORT radiative transfer model (Spurr, 2002) by providing LIDORT with the observation conditions of OMI observations during July 2010, which are used to represent typical viewing conditions of low earth orbit satellite observations, and aerosol profiles from the GEOS-Chem base simulation. To represent typical conditions, these representative scattering weight profiles for each grid box are used to produce the synthetic slant columns."

The difference between using an average scattering weight profile and using individual observation scattering weights is indeed small when averaged over a $4°x5°$ grid. We now discuss this on Line 229:

"Tests performed for all $4°x5°$ grid boxes used here indicate that the mean relative difference between an air mass factor calculated using an average scattering weight profile and the average of air mass factors using observation-specific scattering weight profiles is less than 4%."

**Relevance of prior test cases**

**Of the shape factor test cases described in sect. 3.2.2, the $SF_{trop}$ and $SF_{BL}$ cases are not particularly relevant for satellite measurements. Both of the two main global OMI NO₂ retrievals (NASA SP3, Krotkov et al. 2017; QA4ECV, see Williams et al. 2016 for the NO₂ profiles) use $\sim 1°$ resolution for their NO₂ profiles. Therefore, the $SF_{BL}$ and $SF_{trop}$ cases,**

**which assume one profile globally are not representative of any major satellite product. In fact, the more relevant question, assuming that 4°×5° is still a common resolution for adjoint modeling, is what happens if the satellite prior is *higher* resolution than the model profile.**

**In my opinion, a sixth test case similar to *SF$_{prior}$* but using a set of priors from a 2°×2.5° or 1°×1.25° GEOS-Chem simulation would add value to the paper by studying the effect of having the satellite product's prior at higher spatial resolution than the adjoint model. Also, if the *SF$_{BL}$* and *SF$_{trop}$* cases are retained, it should be clearly stated that they represent extreme cases that do not represent any modern NO$_2$ product.**

Thank you for this suggestion. We now include a test SF$_{finer}$ that uses a set of priors based on a 2°x2.5° GEOS-Chem simulation. We also note in Section 3.2.2 that the SF$_{BL}$ and SF$_{trop}$ cases are extreme cases that do not represent any modern NO$_2$ product (Line 264):

"The *SF$_{BL}$* and *SF$_{trop}$* tests do not represent any modern retrieval algorithms, but are used as extreme examples of using an *a priori* that assumes no spatial variability"

**Other primary concerns**

- **In sect. 3.2, line 210, the authors say that they use one observation per grid box per hour. But OMI will only observe a given location twice per day, maximum, and usually only once per day at about 13:30 local standard time. Are you then filtering these once-per-hour observations down to the ones OMI would actually observe?**

  OMI viewing geometries are used here only as an example of typical viewing geometries of low earth orbit satellite instruments for the scattering weight calculations. The synthetic observations used here are not meant to be synthetic OMI observations or represent the spatial or temporal sampling of OMI. We have clarified this in the text at line 224 as previously noted:

  "…by providing LIDORT with the observation geometry of OMI observations during July 2010, which are used to represent typical viewing geometries of low earth orbit satellite observations, and aerosol profiles from the GEOS-Chem base simulation."

- **I don't follow Eq. (20). Specifically why n$_a$ shows up on the right hand side. Given that:**

$$M(\mathbf{n}) = \frac{\sum_i \mathbf{w}_i \mathbf{n}_i}{\sum_i \mathbf{n}_i}$$

  **and**

$$A_i(\mathbf{n}) = \frac{\mathbf{w}_i}{M(\mathbf{n})}$$

**then to compute $M(\mathbf{n}_m)$ given $A(\mathbf{n}_a)$ you only need to multiply $A(\mathbf{n}_a)$ by $M(\mathbf{n}_a)$ to extract the necessary scattering weights:**

$$M(\mathbf{n}_m) = \frac{\sum_i \mathbf{w}_i \mathbf{n}_{m,i}}{\sum_i \mathbf{n}_{m,i}}$$

$\mathbf{w}_i = A_i(\mathbf{n}_a) \cdot M(\mathbf{n}_a)$

$$\therefore M(\mathbf{n}_m) = \frac{\sum_i A_i(\mathbf{n}_a) M(\mathbf{n}_a) \mathbf{n}_{m,i}}{\sum_i \mathbf{n}_{m,i}}$$

$$= M(\mathbf{n}_a) \frac{\sum_i A_i(\mathbf{n}_a) \mathbf{n}_{m,i}}{\sum_i \mathbf{n}_{m,i}}$$

**I don't think you need $n_a$ to compute $M(\mathbf{n}_m)$ as long as $M(n_a)$ is included in the satellite data (both the NASA SP3 and QA4ECV OMI NO₂ products include the tropospheric AMFs), and given only the AKs and prior profile, it would be difficult if not impossible to compute $M(n_a)$. That means the statement on line 320 about needing the a priori profiles in the dataset is incorrect.**

Thank you for noticing the error in Equation 20. The $n_a$ term on the right hand side should indeed be $n_m$. The text at line 341 has been adjusted accordingly:

"This is most straightforward when scattering weights (rather than averaging kernels) are provided alongside retrieved column data, as scattering weights and shape factors are independently calculated, however simulation-based air mass factors can be calculated using the averaging kernel and *a priori*-based air mass factor via Eq. 19."

**Minor corrections**

- **For Eq. (8), it would be good to make clear that A(z) is the column averaging kernel and therefore a vector, since in Rodgers and Conner, the capital A is typically the full AK matrix. But I agree that following the convention in Eskes and Boersma (2003) is best.**
  We now clarify that A(z) is the column averaging kernel on Line 109.

- **Eq. (9) doesn't seem to be used anywhere else in the paper, and technically is inconsistent with the implicit definition of w(z) in Eq (8). Recommend removing Eq. (9).**

This equation has been removed.

- **In Eq. (17) and Eq. (18) it's unclear what is being summed. Recommend using *I* subscripts to make clear what terms are iterated in the sum.**

This has been changed.

- **On line 198 in sect. 3.1, the authors say that $\Delta$ is computed using either Eq. (12) or (17). Given that much of sect. 2 was spent establishing that these two equations differ, this should be clarified. If I understood correctly, which equation is used effectively depends on which shape factor was used for a given test. If so, I recommend saying that explicitly.**

This is now explicitly stated on line 193: "($\Delta$, from either Eq. (11) if using a simulation-based air mass factor or Eq. (16) if using the retrieval *a priori*-based air mass factor)"

- **For the different shape factors, have you considered the impact of profiles simulated by a model with systematically, rather than randomly (as in $SF_{n30}$), different emissions?**

Thank you for this suggestion. We now include a test $SF_{diffem}$ that considers *a priori* profiles simulated by a model with systematically different emissions.

The mathematical framework outlined in Section 2 presents the main point of the paper, which is that consistency between simulated profiles and shape factor profiles used to calculate AMFs is essential. This is true regardless of model resolution. We now include a test where the *a priori* profile is generated by a higher resolution model (GEOS-Chem run at 2x2.5 resolution). This test further supports the main conclusion of the paper. From Line 316:

[revised manuscript text omitted]
(\boldsymbol{n_m}) = M(\boldsymbol{n_a})\frac{\sum An_{am}}{\sum n_m} \tag{\underline{19}\cancel{20}}$$

It should be noted that both the averaging kernel and scattering weight methods are equivalent for comparisons that examine ratios of retrieved and modeled columns:

$$r_m = \frac{\widehat{\Omega_{v,o}}}{\Omega_{v,m}} = \frac{\Omega_{s,o}\big/M(\boldsymbol{n_m})}{\sum n_m} = \frac{\Omega_{s,o}}{\sum n_m}\frac{\sum n_m}{\sum wn_m} = \frac{\Omega_{s,o}}{\sum wn_m} \tag{20\underline{1}}$$

$$r_a = \frac{\Omega_{v,o}}{\widehat{\Omega_{v,m}}} = \frac{\Omega_{s,o}\big/M(\boldsymbol{n_a})}{\sum An_m} = \frac{\Omega_{s,o}\big/M(\boldsymbol{n_a})}{\sum wn_m/M(\boldsymbol{n_a})} = \frac{\Omega_{s,o}}{\sum wn_m} \tag{21\underline{2}}$$

For ratios, both methods are dependent on geophysical assumptions used to calculate scattering weights but are independent of *a priori* profile information. Lastly, some studies (e.g., Buscela et al., 2013; Qu et al., 2017) may directly assimilate slant column densities rather than vertical
column densities using

$$\Delta_{s,a} = \widehat{\Omega}_{s,m} - \Omega_{s,o} \tag{22}$$

$$= \left( \sum_{i=0}^{tropopause} w_i n_{m,i} \right) - \Omega_{s,o} \tag{23}$$

This approach is also still dependent upon the scattering weights but not upon external *a priori*
profile information. Overall, the choice of approach may be influenced by whether or not
scattering weights are available from either the $NO_2$ retrieval product or radiative transfer
calculations applied to the model. In contrast, use of Eq. (11) or (16) are applicable when these
are not explicitly available or provided.

**3. Tools and Methodology**

**3.1 GEOS-Chem and its adjoint**

The GEOS-Chem chemical transport model (www.geos-chem.org) is used to create
synthetic $NO_2$ observations and for their analysis. The GEOS-Chem version used here is version
35j of the GEOS-Chem Adjoint model. GEOS-Chem includes a detailed oxidant-aerosol
chemical mechanism (Bey et al., 2001; Park et al., 2004) and uses assimilated meteorological
fields from the Goddard Earth Observation System (GEOS-5), with 47 vertical levels up to 0.01
hPa and a horizontal resolution of $4°x5°$. Global anthropogenic $NO_x$ emissions are provided by
the Emission Database for Global Atmospheric Research (EDGAR) inventory (Olivier et al.,
2005) with regional overwrites over North America (EPA/NEI99), Europe (EMEP), Canada
(CAC), Mexico (BRAVO, (Kuhns et al., 2005)), and East Asia (Streets et al., 2006). Other $NO_x$
sources include biomass burning (GFED2 (Van der Werf et al., 2010)), lightning (Murray et al.,
2012), and soils (Wang et al., 1998). This model has been used previously to constrain $NO_x$
emissions (Cooper et al., 2017; Henze et al., 2009; Qu et al., 2017, 2019; Xu et al., 2013; Zhang
et al., 2016).
The GEOS-Chem adjoint (Henze et al., 2007, 2009) is used here to perform a 4D-Var
data assimilation. The adjoint seeks to iteratively minimize a cost function generally defined by
the difference between satellite retrieved and simulated columns (Δ, from either Eq. (11) if using a simulation-based air mass factor or Eq. (16) if using the retrieval *a priori*-based air
mass factor):

$$J = \frac{1}{2}\mathbf{\Delta}^T\mathbf{S}_o^{-1}\mathbf{\Delta} + \frac{1}{2}\gamma_R(\mathbf{E} - \mathbf{E}_a)^T\mathbf{S}_E^{-1}(\mathbf{E} - \mathbf{E}_a)$$  (22)

[revised manuscript text omitted]

---

## Author Response (AR2)

Response to the editor:

**Please clarify whether the response to referees 2's last comment about resolution applies for for priors of much higher resolution, say 2.5km.**

We clarify that finer resolution priors are desirable for retrieval accuracy and that our test is only applicable for inversions at the simulation resolution.

[revised manuscript text omitted]